# Recent Advances in Chemistry of Unsymmetrical Phosphorus-Based Pincer Nickel Complexes: From Design to Catalytic Applications

**DOI:** 10.3390/molecules26134063

**Published:** 2021-07-02

**Authors:** Zufar N. Gafurov, Artyom O. Kantyukov, Alexey A. Kagilev, Alina A. Kagileva, Il’yas F. Sakhapov, Ilya K. Mikhailov, Dmitry G. Yakhvarov

**Affiliations:** 1FRC Kazan Scientific Center, Arbuzov Institute of Organic and Physical Chemistry, Russian Academy of Sciences, 420088 Kazan, Russia; kant.art@mail.ru (A.O.K.); al-kagilev@mail.ru (A.A.K.); vtb241997@mail.ru (A.A.K.); sakhapovilyas@mail.ru (I.F.S.); 2Alexander Butlerov Institute of Chemistry, Kazan Federal University, 420008 Kazan, Russia; tiimhailovilya@gmail.com

**Keywords:** pincer ligands, unsymmetrical ligands, nickel complexes, PCN and POCN complexes, NNP and PNP complexes, NHC complexes, catalysts

## Abstract

Pincer complexes play an important role in organometallic chemistry; in particular, their use as homogeneous catalysts for organic transformations has increased dramatically in recent years. The high catalytic activity of such bis-cyclometallic complexes is associated with the easy tunability of their properties. Moreover, the phosphorus-based unsymmetrical pincers showed higher catalytic activity than the corresponding symmetrical analogues in several catalytic reactions. However, in modern literature, an increasing interest in the development of catalysts based on non-precious metals is observed. For example, nickel, which is an affordable and sustainable analogue of platinum and palladium, known for its low toxicity, has attracted increasing attention in the catalytic chemistry of transition metals in recent years. Thus, this mini-review is devoted to the recent advances in the chemistry of unsymmetrical phosphorus-based pincer nickel complexes, including the ligand design, the synthesis of nickel complexes and their catalytic applications.

## 1. Introduction

The control of the properties of metal complexes by varying the nature of the ligand to obtain highly efficient catalytic systems and materials with useful properties is one of the main tasks of modern organometallic and inorganic chemistry. Fine-tuning of the steric and electronic parameters of the formed metal complexes can be realized by chelation by binding the ligand to the metal through two or more bonds, which leads to a dramatic change in the properties of metal complexes such as stability and reactivity [1,2].

The introduction of a direct metal–carbon bond in chelate systems leads to the formation of an organometallic complex with a metal cycle, providing additional stabilization of this bond [3,4,5,6,7,8,9,10,11,12,13,14,15,16,17,18,19]. Pincer ligands are tridentate and typically bind to a metal in a meridional fashion, forming complexes of general formula [M(E^1^YE^2^)L_m_]X_n_ (Figure 1), where E^1^ and E^2^ are typically a neutral two-electron donor group (such as amine or phosphine group), while Y is usually represented by an anionic carbon atom of a 2,6-disubstituted aryl moiety or, less often, pyridine group; X and L are counterions and auxiliary ligands, respectively; M is metal [20]. Pincer complexes play an important role in organometallic chemistry mainly due to their versatility, arising from the numerous possibilities to adjust the ligand environment, as well as due to their high thermal stability, which is mainly attributed to the rigidity of the pincer framework [2]. In particular, their use as homogeneous catalysts for organic transformations has increased dramatically in recent years [1,3,4,12,13,14,15]. The high catalytic activity of such bis-cyclometallic complexes is associated with the easy tunability of their properties [8]. PCP-type nickel complexes have been reported as the first example of nickel pincer systems [21], followed by NCN pincer nickel complexes [22,23]; this success promoted the development of the chemistry of pincer ligands as a whole. These two types of ligands show different influence of the properties of the formed nickel complexes. For example, the use of an NCN pincer ligand enabled the stabilization of nickel (III) complexes, while PCP pincer nickel complexes engaged in the activation of small molecules. These early pincer ligands were symmetric with respect to ligand “arms” [21,24,25,26,27,28,29,30,31,32,33,34].

However an increasing interest in the development of nickel complexes based on unsymmetrical phosphorus containing pincer ligands (PCN, POCN, PNN, PCO, PCS, etc.) is observed in modern literature [35,36,37,38,39,40,41,42,43,44].

Thus, metal complexes based on pincer ligands have established themselves as highly effective catalysts for such processes as the formation of a new carbon–carbon bond (Heck, Suzuki–Miyaura reactions), hydrogenation of aryl and alkyl ketones, alkane dehydrogenation, hydrosilylation, oligo- and polymerization of unsaturated hydrocarbons and others [8]; moreover, unsymmetrical pincer complexes showed higher catalytic activity than their symmetrical analogues [45,46,47,48]. Thus, Milshtein and colleagues described the better catalytic activity of unsymmetrical ruthenium pincer complexes for the dehydrogenation of primary alcohols to esters and hydrogen [49] and hydrogenation of esters to alcohols [40,50]. Gebbink, Szabó et al. showed higher TOF values for the unsymmetrical palladium PCS complex compared to PCP and SCS complexes in aldol reactions [44]. Huang and colleagues reported that the benzoquinoline-based iridium PCN complex catalyzes the alkene hydrogenation reaction more efficiently than the PCP counterpart [51]. Shubina et al. demonstrated the superior catalytic activity of the PCN pincer complex of iridium containing a pyrazole N-donor compared to the PCP complex in the dehydrogenation of amine boranes [52].

The chemistry of pincer compounds, including their catalytic use, is described in detail in a number of works [3,13,14,15,28,53,54,55,56,57,58,59,60,61,62,63]. Most of them are devoted to the complexes based on metals such as ruthenium, rhodium, iridium, palladium and platinum. However, in modern literature, there is an increasing interest in the development of catalysts based on non-precious metals. For example, nickel, which is a more affordable low-cost and high-natural-abundance analogue of platinum and palladium, has attracted increasing attention in the catalytic chemistry of transition metals in recent years [1,64,65,66,67,68,69,70,71,72,73]. Thus, this mini-review is devoted to the recent advances in the chemistry of unsymmetrical phosphorus-based pincer nickel complexes, including the ligand design, the synthesis of nickel complexes and their catalytic applications.

## 2. Synthesis of Nickel Pincer Complexes

The methods most widely used for the synthesis of nickel pincer complexes include: cyclometalation with activation of C-H bond, oxidative addition and transmetalation (Scheme 1). Obviously, cyclometalation is the most convenient method since it does not require an additional prefunctionalization of the ligand and avoids using low-stable and flammable precursors [1].

PCP-containing metallacycles are the most common representatives of nickel pincer complexes. The first example of a PCP nickel complex is {2,6-Bis[(di-tert-butylphosphino)methyl]phenyl}chloronickel (**1**), obtained in 1976 [21], followed by NCN pincer nickel complexes **2a** and **2b** (Scheme 2) [22,23,74,75].

Based on the differences in the electronic and steric properties of the PCP- and NCN-type ligands, the preparation of the corresponding nickel complexes is based on different strategies. Nickel PCP complexes can be obtained by direct C-H bond activation of the corresponding PCP ligand, while the use of Ni(0) precursor [Ni(COD)_2_] or the preliminary lithiation of the NCN ligand followed by the replacement of lithium with nickel is a necessary condition for obtaining nickel NCN complexes. Important differences between these types of complexes are manifested in both stoichiometric and catalytic reactions. For example, the use of the pincer NCN ligands made it possible to stabilize and isolate corresponding nickel (III) species and significantly increase the catalytic activity of divalent nickel halide complexes in the addition of polyhalogenated hydrocarbons to olefins (Kharasch addition) [76], while PCP nickel complexes are involved in the activation of small molecules due to the strong coordination of the ligand (Figure 2) [30,33,77]. Thus, various symmetric pincer nickel complexes were synthesized and used in homogeneous catalysis.

## 3. POCN Complexes

Unsymmetrical nickel pincer complexes are less studied, and one of the first examples was represented by the POCN-type phosphinite amine ligands (complexes **3a**–**3c** in Scheme 3; see Table A1 in Appendix A for selected bond lengths (Å) of unsymmetrical nickel pincer complexes presented in this paper) described by Zargarian and Miller [78,79,80,81]. It is interesting to note that, unlike their symmetric NCN analogues, these complexes can be obtained by direct metalation with activation of the C-H bond, and, unlike PCP analogues, they can be oxidized with bromine to form stable nickel (III) complexes (complexes **4a**–**4c** in Scheme 3). Thus, POCN nickel complexes combine the unique properties of both classes of ligands. However, the synthesis of unsymmetrical pincer ligands is more complex and requires multiple steps (Scheme 3).

The presence of bulky substituents at the phosphorus atom can increase the stability of the resulting nickel complexes and allow their catalytically active forms to be isolated. Therefore, the same authors obtained a complex containing a secondary amine in its structure (complex **5** in Figure 3). This complex is capable of dimerization with the formation of binuclear complex **6**, which turned out to be an active catalyst in the processes of functionalization of acrylonitrile in alcoholic media [79].

Since then, many other nickel complexes containing unsymmetrical POCN ligands have been described [82], including various substituted analogues of complex **3a**, such as compound **7** (Figure 4) [80]. Optically active species are also known, for example, a complex containing the imidazole fragment (**8** and **9**) (Figure 4), which is successfully used in the asymmetric Suzuki–Miyaura cross-coupling reaction [83,84,85,86], as well as complex **10** [87], containing diisopropylphosphinite and imine fragments, which can be oxidized with bromine or *N*-bromosuccinimide to complex **11** (Figure 4). It is interesting to note that complex **12** also obtained by Zargarian and co-workers, containing the diphenylphosphinite fragment, undergoes two-electron oxidation of the phosphinite fragment with the formation of complex **13** upon interaction with bromine (Figure 4). Thus, we can conclude that the choice of the P-substituent is very important for the stabilization of Ni (III) particles [87].

## 4. PCN Complexes

Phosphorus-based complexes such as PCN-type systems are particularly intriguing since, unlike their symmetric NCN analogues, their nickel complexes can be obtained by direct cyclometalation with activation of the C-H bond, and, unlike PCP analogues, they can stabilize nickel (III) species. Thus, PCN nickel complexes combine the unique properties of both classes of N- and P-based ligands [88,89,90,91,92,93,94]. In such species, the difference in the trans effect between the two different donor arms provides the hemilability of the ligand since the group with the weaker trans effect (N) is more likely to dissociate from the metal center, which leads to a vacant coordination site at the metal center and thus can allow for effective coordination and transformation of substrate molecules in the homogeneous catalysis conditions [95].

Recently, Moussa and co-authors reported the synthesis of PCN complexes of nickel with ditertbutylphosphine and dimethylamine fragments (Scheme 4) [96]. It is interesting to note that the cyclometalation of this PCN ligand with anhydrous nickel chloride in toluene in the presence of 4-dimethylaminopyridine (DMAP) leads to the formation of complex **14a** with 63% yield, while this reaction in THF in the presence of triethylamine allows complex **14a** to be obtained in 86.5% yield. Such a low yield of compound **14a** in the first case is associated with the formation of a paramagnetic complex **15** as a by-product. Triethylamine is a stronger base than DMAP, which avoids protonation of the amine fragment of the PCN ligand with acid (HCl), formed during the reaction. The bromide analogue **14b** was obtained by a similar method with a yield of 96%.

Interestingly, the P-Ni bond in **14a** is significantly shorter, than the corresponding bond in its symmetrical PCP analogue (complex **1**); at the same time, the N-Ni bond is found to be much longer than in symmetrical NCN complex **2b** (Table 1). These differences are due to the strong trans influence of the phosphine moiety, and they determine hemilability.

On the cyclic voltammograms of complexes **14a** and **14b** in dichloromethane, irreversible oxidation peaks at E_1/2_ = 0.837 and 0.797 V for **14a** and **14b**, respectively, are observed [96]. These oxidation potentials are lower than the corresponding values for complexes of nickel with POCN ligands (around 1.0 V), which is associated with a higher donor ability of PCN ligand. Thus, PCN nickel (III) complexes can be easily obtained by oxidation of the corresponding nickel (II) complexes. Indeed, the interaction of complexes **14a** and **14b** with anhydrous salts of CuX_2_ (X = Cl, Br) in dichloromethane leads to the formation of the corresponding Ni(III) complexes **16a** and **16b** (Scheme 5) [96]. Both of the obtained complexes in the crystal display distorted square-pyramidal coordination geometries around nickel with a halide in the apical position.

It is well known that trivalent nickel complexes are important intermediates of the Kharasch addition reaction [97]. Therefore, NCN nickel complexes are efficient for this process, while the aromatic PCP nickel pincer analogues are not suitable as catalysts in this reaction mainly because they do not produce the corresponding Ni(III) species. PCN complexes **14a** and **14b** were tested in this process, showing higher catalytic activity in the process of addition of CCl_4_ to styrene than their POCN analogues; however, they turned out to be less active than NCN nickel complexes. Thus, the catalytic efficiency of nickel pincer complexes in the Kharasch addition reaction appears in the following order: NCN > POC_sp3_OP > PCN > POCN [78].

It is also worth noting that the interaction of **14a** and **14b** with Grignard reagents yielded the first examples of alkyl and aryl-nickel complexes based on an unsymmetrical aromatic pincer ligand **17a** and **17b** (Scheme 6) [96].

Later, Moussa and co-authors presented the synthesis of an unsymmetrical PCN pincer complex of nickel bearing tertbutyl groups on the phosphorus atom and isopropyl groups on the nitrogen donor atom (complex **18** in Scheme 7) [98]. The corresponding methyl complex **19** was also obtained from **18** through a transmetalation reaction with MeMgCl. Complex **19** demonstrated a high reactivity in a carboxylation of a Ni−C bond using CO_2_ with the formation of acetate complex **20**.

In 2019, Luconi et al. demonstrated a synthesis of PCN nickel complexes containing ditertbutylphosphine and pyrazolyl fragments and different halogens in the structure (F, Cl, Br, I; see Scheme 8) [88]. The electrochemical properties of the obtained complexes **21a**-**21d** were investigated using cyclic voltammetry and in situ EPR spectroelectrochemistry.

Authors found that the complexes exhibit different behaviors during electro-oxidation. Thus, complexes **21a** and **21d** during the electro-oxidation process (proceeding at E_1/2_ = 0.44 and 0.29 V for **21a** and **21d**, respectively) form halogen-free species of nickel (III), while chloride and bromide complexes **21b** and **21c** (E_1/2_ = 0.79 and 0.83 V, respectively) change the square-planar geometry of the molecule to a distorted tetrahedron, preserving the halogen atom to the coordination sphere of nickel (Scheme 9), as evidenced by the hyperfine interactions on their EPR spectra [88].

Authors have also found that the high nucleophilicity of the fluoride ligand in **21a** leads to halogen bonding with the electrophilic iodine atom in iodopentafluorobenzene (C_6_F_5_I). However, the halogen-bonding capability is weaker than that of the symmetrical analogue (^tBu^PCP)NiF [^tBu^PCP = 2,6-C_6_H_3_(CH_2_P*^t^*Bu_2_)_2_], and it is featured by a positive ΔS° value because of a higher degree of aggregation in solution [88,99].

Interestingly, the oxidation of complexes **2****1b** and **21c** with anhydrous CuX_2_ salts (X = Cl, Br) did not lead to stable Ni(III) species, which were observed only in situ by EPR spectroscopy [89]. The presence of the aromatic pyrazole ligand as a weaker sigma donor compared with an aliphatic -NR_2_ group (complexes such **3a**–**3c**, **9**, **14a** and **14b**) is detrimental for the stabilization of the electron-poorer Ni(III) complexes.

The interaction of chloride complex **21b** with an excess of sodium borohydride led to the formation of the relatively robust borohydride complex **22** (Scheme 10) [89], which was isolated and characterized by X-ray diffraction. Interestingly, the reaction of symmetrical (^tBu^PCP)NiCl with NaBH_4_ generates (^tBu^PCP)NiH as the isolable product with (^tBu^PCP)Ni(BH_4_) observed as intermediate in solution by NMR spectroscopy [99] and only later was isolated by Enthaler and co-workers [100].

The nickel complexes **21a**–**21d** preactivated by MMAO have shown moderate activity in the ethylene oligomerization process leading to even-numbered olefins, predominantly C_4_-C_10_ fractions. To the best of our knowledge, this is the first example of application of PCN nickel complexes on the ethylene oligomerization reaction [101,102,103,104,105,106].

It also should be mentioned that Luconi and co-authors demonstrated the synthesis of unusual benzothiazole-based unsymmetrical pincer nickel complexes **23a**, **23b** and **24** (Scheme 11) [107]. The reaction of the corresponding benzothiazole-based pincer ligand with nickel dibromide in toluene in the presence of Et_3_N as a base leads to the bromide complex **23a**. Fluoride analogue **23b** was obtained by treatment of **23a** with silver fluoride in toluene (Scheme 11). The ionic *aquo* complex **24** was synthesized through bromide abstraction from **23a** with a silver tetrafluoroborate in THF. The water molecule came from the AgBF_4_ or from THF. The N atom in the thiazole ring is a stronger donor, therefore no sulfur coordination was ever observed. Interestingly, binding of ligand to a metal ion generates coordination rings of different sizes: a five-membered cycle on the phosphine arm and a six-membered cycle on the benzothiazole arm. Moreover, the oxo-bridge induces a distortion of the ligand, which is not lying on the P-Ni-N plane (Figure 5).

Table 2 represents the comparison of P-Ni and N-Ni bond distances of analogous PCN nickel pincers with different N-donor groups (dimethylamine for **14a**, pyrazole for **21c** and benzothiazole for **23a**). The corresponding distances of **21c** and **23a** are almost equal due to the similar basicity of N atom in the thiazole and pyrazole rings (*pK_a_* of the conjugate acid 2.50 vs. 2.48 for thiazole and pyrazole, respectively), while a higher *pK_a_* value of the dimethylamine (10.73) indicates stronger basicity. Since lower *pK_a_* values of electron-poor pyrazole and benzothiazole groups indicate their weaker σ-donor ability, these groups more likely dissociate from the metal center, yielding more pronounced hemilability, which leads to higher catalytic activity of related complexes in various catalytic processes compared to dimethylamine-containing analogues.

Interestingly, complexes **23a** and **23b** preactivated by MMAO demonstrated high catalytic activity in ethylene oligomerization (up to 200 × 10^3^ mol_C2H4_∙mol_Ni_^−1^∙h^−1^) with the formation of even-numbered olefins (mainly C_4_-C_10_ fractions) as products. The comparison of their performance with the results obtained for more rigid pyrazole-based analogues **21a** and **21c** demonstrates a positive effect of the flexibility modification of the ligand [108].

Recently, Zargarian and co-workers demonstrated the synthesis of unsymmetrical pincer Ni-C_sp_^3^ complexes **25a–25f** containing two metallacycles of different sizes: a six-membered cycle on the phosphine side and a four-membered cycle on the amine side (Scheme 12). The synthetic methodology presented in this study is based on the reactivity of phosphinite derived from 2-vinylphenol. Starting phosphinite, nickel (II) precursor and a base (Et_3_N) in THF at room temperature produce the target complexes. The solid-state structure of the complexes was established by XRD analyses, which revealed that the Ni-P bond is typically longer than the corresponding Ni-N bond (Figure 6). Interestingly, authors observed isomerization of these complexes due to the hemilability of the N-Ni coordination bond, which may provide the basis for the future rational design of new active systems, since this hemilability generates a vacant coordination site, which is a prerequisite for the obtainment of an active catalyst. Moreover, CV experiments revealed that these complexes undergo facile one-electron oxidation due to the impact of a C_sp_^3^−Ni moiety [109].

## 5. NNP Complexes

The other important class of phosphorus and nitrogen containing unsymmetrical pincer nickel systems is NNP complexes. The combination of pincer-type ligands with a nickel ion has attracted much attention to create effective cross-coupling catalysts in recent decades. Mostly, aryl chlorides, bromides and iodides are used as electrophiles in these processes [110,111], while the activation of aryl fluorides using nickel catalysts is still a challenge in synthetic chemistry [112,113]. In 2019, Yamaguchi and co-workers described the synthesis of β-aminoketonato- and β-diketiminato-based unsymmetrical pincer complexes **26**–**28** (Scheme 13), whose catalytic performances in the cross-coupling of aryl chlorides with aryl Grignard reagents [114] and in the highly Markovnikov-selective hydroboration of vinylarenes using bis(pinacolato)diboron have been investigated [115]. It was found that the presence of the phosphorus group in complex **27** brings to remarkable catalytic activity in the cross-coupling process over the phosphorus-free analogues, while the β-diketiminato framework increases the energy level of the HOMO of the complex **28,** according to the DFT study, making the system electronically more favorable. However, **NNN** complex **28** is not an effective catalyst for cross-coupling reactions. Later, in 2019, the working group combined these features introducing a phosphorus donor into the β-diketiminato framework to create a sterically and electronically favorable environment at the Ni center (complex **29**), which was prepared by the reaction of the nickel (II) precursor [NiCl_2_(2,4-lutidine)_2_] with the lithiated NNP ligand and effectively facilitated the cross-coupling of aryl fluorides with aryl Grignard reagents (Table 3) [116].

The reaction was carried out with 1.0 mmol of fluoroarene and 1.2 mmol of arylmagnesium bromide in the presence of a Ni (II) complex (0.05 mmol) in THF (5 mL) at room temperature for 6 h [116].

Shortly afterward, the synthesis of the first pyrrole-based unsymmetrical NNP pincer complexes of nickel (**30a**–**30c**) containing diphenylphosphine and pyrazole fragments was presented (Scheme 14) [117]. 1:1 reaction between NNP ligand and [NiCl_2_(DME)] or NiX_2_ (X = Br, I) in acetonitrile in the presence of NEt_3_ under reflux conditions afforded the pincer nickel complexes **30a**–**30c** with good yields. The pyrrole-based pincers are well known to stabilize a variety of nickel complexes [118,119,120]. Authors compared the efficiencies of complexes **30a**–**30c** in catalyzing norbornene to polynorbornene with efficiencies of their symmetrical analogues—PNP complexes **32a**–**32c** (Scheme 14) [121]—which were synthesized by the reaction between PNP pincer ligand and [Ni(OAc)_2_]·4H_2_O with the formation of **31**, which further treated with an excess of LiCl or LiBr or KI in acetone/water at room temperature to give the corresponding halide ion substituted Ni complexes **32a**–**32c**.

Interestingly, according to X-ray diffraction analysis the Ni-P distance in **30b** is 2.1613 Å, while for **30c**, it is 2.1647 Å, which are shorter than the distances found in their symmetrical PNP analogues **32b** and **32c**, indicating the strong trans effect of the P atom in comparison to the N atom (Figure 7). Moreover, the pyrrolide N−Ni distances in **30b** and **30c** (1.886 Å and 1.847 Å, correspondingly) are shorter than the amide N−Ni distances found in the analogous pincer nickel (II) complexes: 1.924 Å in [NiCl{N(SiMe_2_CH_2_PPh_2_)_2_}] [122] and 1.895/1.912 Å in [NiX{N(*o*-C_6_H_4_PPh_2_)_2_}] (X = Cl, Br) [110], which indicate the strong bond formed by the pyrrolide N atom.

The authors carried out a systematic study of the polymerization of norbornene using symmetric (**32a**–**32c**) and unsymmetrical (**30a**–**30c**) pyrrole-based complexes. Nickel pincer NNP complexes in the presence of MMAO or EtAlCl_2_ showed high yields and high activities (in the range of 10^7^ g of PNB mol^−1^ h^−1^). Meanwhile, symmetric PNP complexes, on the contrary, turned out to be ineffective catalysts in this process, suggesting that with an increase in the number of N donors, the yield and activity of the catalytic system increase, following the order PNN > PNP (Table 4).

It is interesting to note that nickel complexes **33a**–**33d** with diarylamido-based unsymmetrical pincer ligands containing a chiral oxazoline ring (Scheme 15) showed much lower activity during the catalytic polymerization of norbornene in the presence of MAO (in the range 0.12–1.5 × 10^5^ g of PNB mol^−1^ h^−1^) [123]. Meanwhile, the analogous palladium complexes exhibited relatively higher activities (in the range of 4 × 10^6^ g of PNB mol^−1^ h^−1^). However, it has been demonstrated that the steric hindrance and electronic effect of the ligands have a significant influence on the complexes and the consequent catalytic properties (the activity increase, following the order **33c** < **33b** < **33d** < **33a**, see Table 5).

Recently, Gardinier and Wang demonstrated synthesis of unsymmetrical PNN pincer nickel complexes **34a**−**34c** with pyrazolyl and diphenylphosphino donor groups (Scheme 16) [124]. These complexes were tested in hydrodehalogenation reactions of 1-bromooctane and different aryl halides in using sodium borohydride as a hydride source and as a base. **34a** derivative showed the best catalytic performance that correlates with its electron donor properties [125].

One of the latest published *cationic* unsymmetrical pincer complexes of nickel was synthesized by the reaction of anhydrous NiCl_2_ with 1.2 equivalents of the Milstein’s NNP ligand in THF (Scheme 17) [126,127]. In **35**, the Ni−N(pyridyl) distance (1.876 Å) is shorter than the Ni−N(amine) bond length (2.014 Å). The complex was tested as a catalyst precursor for the hydrogenation of CO_2_ to formamide in anhydrous DMSO as a solvent; however, no catalytic activity was observed in the presence of morpholine or dimethylamine as substrates under the applied reaction conditions.

Analogously to **14a**, the Ni-P bond in **35** is significantly shorter than the corresponding bond in its symmetrical PNP analogue [128]; at the same time, the Ni-N(amine) bond of **35** is much longer than in the symmetrical NNN complex (Figure 8) [129]. These differences are due to the strong *trans* influence of the phosphine moiety, and they determine hemilability (see Table A1 in Appendix A for selected bond lengths (Å) of unsymmetrical nickel pincer complexes presented in this paper).

## 6. PNP’ Complexes

The unsymmetrical ^R^PNP^R′^ pincer hydride nickel complexes (**36a**–**36c**) containing various substituents at the phosphorus atom (R = Ph, R′ = ^i^Pr, Cy) are also known in the literature. These complexes can be obtained by direct interaction of the corresponding ^R^PNP^R′^ ligand with the zero-valent metal precursor Ni(COD)_2_ with a yield of 84–89% (Scheme 18) [130,131].

It was found that these complexes are capable of activating small molecules, such as O_2_, CO, olefins and some others [130,131,132,133]. The reaction includes the insertion of the substrate into the Ni-H bond. For example, complex **36a** reacts with olefins such as ethylene, 1-hexene, norbornene and styrene to generate complexes **37a**–**37c**, **38** as thermally stable solids (Scheme 19) [131]. In contrast to the selective 1,2-insertion found for 1-hexene, styrene inserts into the Ni-H bond of **36a** in an exclusively 2,1-manner. In addition, **36a** reacts with electronically activated olefins such as methyl acrylate affording product **39**.

It should be noted that authors described the reactivity of obtained complexes **37a**–**37c** toward carbon monoxide. It was found, that migratory insertion of CO into the Ni-R bonds affords Ni(II)-acyl derivatives **40a**–**40c** (Scheme 20) [133].

Interestingly, further carbonylation of acyl compound **40a** leads to generation of nickel zero complex **41** as a result of the C−N bond-forming reductive elimination whereas no reaction occurs for 40**b** and **36c** under similar conditions. At the same time, in the presence of carbon monoxide, hydride complex **34** undergoes exclusively N−H bond-forming reductive elimination to generate zero-valent nickel dicarbonyl derivatives.

## 7. *N*-Heterocyclic Carbene Complexes

*N*-Heterocyclic carbenes (NHCs) are isolobal with electron-rich phosphines and they are often considered as phosphine ligand alternatives [134]; therefore, unsymmetrical pincer nickel complexes bearing NHC groups are also discussed herein. These carbenes show low toxicity and strong σ-donating properties easily tunable by varying the substituents at the nitrogen atom and known to exert electronic and steric effects. Due to these unique properties, NHCs are a versatile and indispensable class of ligands applied in coordination chemistry and homogeneous catalysis by transition metal complexes. Consequently, NHC-based C donors have been commonly employed in symmetrical-type ligands. NHC-containing unsymmetrical pincer-type ligands have also been reported by adding different donors to the pincer architectures.

As a very good example of unsymmetrical pincer complexes of nickel with NHC-moiety, Sun and co-workers reported the preparation and characterization of the CNN-pincer complex **42a**–**42c** (Scheme 21) [135]. Complexes were synthesized via the lithiation of related pincer ligands and transmetalation by Ni(DME)Br_2_ precursor.

Obtained complexes showed high catalytic activity in Kumada cross-coupling reactions of aryl chlorides or dichlorides with aryl Grignard reagents under mild conditions as well as in Sonogashira coupling reactions of aliphatic halides with terminal alkynes [136].

It has also been confirmed that the hydrogenation of nickel halides **42a**–**42c** in the presence of (EtO)_3_SiH and NaO*^t^*Bu as a base leads to the formation of hydrido nickel complexes **43a**–**43c** (Scheme 22) [137]. According to the X-ray diffraction analysis of **42a** and **43b**, the Ni−N(amine) bond distance is remarkably longer than the Ni−N(amide) due to the strong trans influence of the carbon atom of the NHC coordination moiety. Complex **43a** exhibited high catalytic activity in the hydrodehalogenation of organic halides.

Later, authors introduced a pyrrolidine group, instead of a dimethyl amino group to the pincer ligand structure and synthesized complexes **44a** and **44b** by the direct cyclometalation of the corresponding ligand precursor in the presence of Et_3_N (Scheme 23) [138]. Interestingly, the yields of **44a** and **44b** were higher (85 and 90%, respectively) than the yields of **42a** and **42b** (80–87%).

Complexes **42a**, **42b**, **44a** and **44b** were examined in transfer hydrogenation of ketones in the presence of a base (NaO*^t^*Bu or KO*^t^*Bu). The results show that the catalytic activity follows the order **44a** > **42a** > **44b** > **42b** for acetophenone hydrogenation. Authors considered that the strong electron-donating properties of the pyrrolidine and iso-propyl group in complex **44a** made the metal center electron-rich. Authors also proposed the hydrido nickel complex to be the intermediate of this process, whose electron-rich property would increase the activity of the Ni-H bond. Thus, hydrido complex **43a** was tested as the catalyst to explore the mechanism of the interaction. Indeed, it was found that complexes **42a**, **42b**, **44a** and **44b** were catalyst precursors, which generate the corresponding catalytically active hydrido complexes.

It is also interesting to note the synthesis of the NHC pincer complexes of nickel with both unsymmetrical and chiral properties. The cyclometalation was carried out via direct C−H bond activation in the presence of NaOAc to form corresponding complexes **45a**–**45f** (Scheme 24) [139]. However, the yields of **45a**–**45f** were in the range of 15–27%, and they did not find their place in catalysis yet. However, their palladium analogues showed moderate stereoselectivities in the asymmetric Friedel−Crafts alkylation.

It is also worth mentioning the unsymmetrical nickel complex with NHC-triazole arms, synthesized by the reaction of a ligand precursor with Ag_2_O with in situ formation of a silver NHC complex, which was transmetalated by [NiCl_2_(PPh_3_)_2_] to provide desired complex **46** (Scheme 25) with 32% yield [140]. Complex **46** was found to be highly active in the Suzuki–Miyaura cross-coupling reactions of aryl bromides or aryl iodides with phenylboronic acid at 110 °C.

The electrochemical generation of NHC containing unsymmetrical pincer nickel complex **47** has been described by some of us (Scheme 26) [141,142]. It was found that cathodic reduction of the imidazolium salt (LHBr in Scheme 26) in the presence of nickel (II) ions leads to the formation of *N*-heterocyclic carbene complexes of nickel (II) with the molecular hydrogen as the only byproduct. Interestingly, the imidazolium salt plays a role of both NHC source and electrolyte, while the source of nickel ions is the nickel plate.

Vabre et al. demonstrated the synthesis of nickel complexes with PIMCOP, PIMIOCOP and NHCCOP ligands (Scheme 27) [143]. PIMCOP complex **48** was obtained by nickelation of an unsymmetrical meta-phenylene-based ligand bearing phosphinite and imidazolophosphine groups. *N*-methylation of the latter generates a new PIMIOCOP complex **49** with NHC moiety, which can be converted subsequently into NHCCOP complex **50** (Scheme 27). Interestingly, the Ni-P distance in **50** is much longer (2.145 Å) compared to the related Ni-P bond of **48** and **49** (2.132 and 2.133 Å). These indicate the much stronger trans influence of the NHC moiety compared to the phosphine group.

## 8. Other Types of Complexes

Other types of unsymmetrical pincer nickel complexes are also known in the literature. For example, phosphinito-thiophosphinito PSCOP complex **51** (Scheme 28), which was obtained by cyclometalation of the related ligand by NiCl_2_ [144]. The X-ray diffraction analysis showed that the nickel center is located in a slightly distorted square-planar geometry. PSCOP complex **51** demonstrated a good catalytic activity and selectivity in C-S couplings of iodobenzene with different disulfides.

Other sulfur-containing nickel complexes are SCS-type pincers. Interestingly, SCS-ligand-based nickel pincer nucleotide (lactate racemase) is the first enzyme known to contain a pincer metal complex as a cofactor [145]. This enzyme converts L-lactic acid into D-lactic acid. The mechanisms of this transformation have been studied using DFT calculations. The most probable pathway for proton-coupled hydride transfer involves the intermediate formation of an unsymmetrical complex [146]. Inspired by the structures of the active site of lactate racemase, similar scorpion-like SCS nickel complexes were proposed, and their potentials for catalytic hydrogenation of CO_2_ [147] and for the asymmetric transfer hydrogenation of 1-acetonaphthone (the phosphorus-based complexes **52a**–**52d** in Figure 9) [148] were computationally predicted.

It is also interesting to mention the phosphinite PONNOP expanded pincer ligand, which undergoes an unexpected rearrangement in the presence of nickel (II) species to form an unusual PONNP complex **53** (Scheme 29) [149]. Interestingly, the reaction of the PONNOP ligand with copper chloride did not lead to the rearrangement but to a binuclear PONNOP copper complex. The performed DFT calculations confirmed that PONNOP and PONNP free ligand isomers are similar in energy and have a high barrier for their interconversion. However, authors concluded that the P atom, which migrates from O to N, is bound to the transition metal atom, which provides the additional stabilization of the transition state to enable the isomerization toward the more stable PONNP pincer motif.

More recently, the synthesis of a series of unsymmetrical Ni(II)-POCOP pincer complexes (**54a**–**54c**) derived from 1,7-naphthalenediol were described (Scheme 30) [150]. The molecular structures of two complexes were determined by single-crystal X-ray diffraction analysis. Interestingly, the Ni-P distances were almost identical, despite the size of the metallocycle. For example, for **54a**, the M-P^1^ bond length was 2.1340 Å, while that for the M-P^2^ distance was slightly longer (2.1417 Å). These complexes were examined as catalysts in Suzuki–Miyaura C-C cross-coupling reactions. However, the results are not as good as those obtained with their analogous POCOP pincer derivatives of 1,3-naphtalene diol [151]. Complex **54c**, bearing a more sterically demanding – PPh_2_ group, was the most efficient catalyst of the series, reaching 53% yield of the biphenyl product.

Very recently, Wang et al. demonstrated the synthesis of unsymmetrical PCP′ pincer nickel complexes bearing a phosphine-phosphinite ligand with a stereogenic carbon center [152]. The corresponding enantiopure nickel complexes **55a** and **55b** were obtained via a convenient one-pot three-step reaction (Scheme 31). Moreover, the authors showed the possibility of the replacement of the chloride ligand in **55b** by the action of silver agents AgOAc or AgBF_4_ with the formation of a neutral complex **56b** or a cationic complex **57b** (Scheme 31). The crystal structures of complexes **55a** and **56b** were determined by X-ray diffraction analysis (Figure 10), which revealed that, in **55a**, atoms of phosphorus lie on the same plane defined by the [PCP’Cl] atoms. Meanwhile, 3,5-methyl substituents and –OAc group in **56b** lead to the distortion of the ligand in which the phosphine moiety is not located in the same plane [PCP’Cl].

Authors applied these newly developed nickel pincer complexes for the highly enantioselective catalytic synthesis of P-stereogenic secondary phosphine-boranes by asymmetric addition of primary phosphine to electron-deficient alkenes. The resulting products may be easily converted into useful chiral phosphine ligands for asymmetric transformations. Interestingly, the nickel-based catalysts demonstrated the improved enantio- and diastereoselectivity to 94% ee and 13:1 dr (for complex **55b**, entry 2) compared to palladium-based analogues (59% ee with low dr). While Ni-OAc complex **56b** exhibited similar catalytic activity compared to **55b** (entry 3, Table 6), in contrast, the cationic complex **57b** did not catalyze the reaction at all (entry 4, Table 6). Moreover, authors revealed that CH_2_Cl_2_ as a solvent and temperature of −40 °C is the best combination (entries 8–11, Table 6) compared to other conditions (entries 5–7, Table 6). Control experiments confirmed that both the catalyst and the base were necessary for product formation (entries 10 and 11, Table 6).

According to the proposed by the authors catalytic cycle for **55b**, catalyzed synthesis of secondary phosphine-boranes includes the formation of Ni-OAc (**56b**) complex by the ligand exchange reaction, followed by the transphosphination step between diphenylphosphine and formed complex, which leads to a nickel phosphide intermediate. Then, the nucleophilic phosphorus addition to the β-position of enone affords a nickel phosphine complex, which, finally, reacts with HOAc, formed at the transphosphination step, and releases the product as well as regenerates the active catalyst.

## 9. Conclusions

In summary, the chemistry of unsymmetrical P-based nickel pincer complexes is young and started its way only in 2008, which is 32 years later after the synthesis of the first symmetrical nickel pincer complex (Figure 11). However, the number of publications in this area has increased dramatically in recent years. Research gains new unprecedented complexes with unexpected properties in the past few years. The development in this field seems to be very promising [109,124,125,153,154].

As follows from the review, unlike their symmetric NCN analogues, unsymmetrical PCN complexes can be obtained by direct metalation with activation of the C-H bond, and, unlike PCP analogues, they can be oxidized with bromine to form stable nickel (III) complexes. Thus, unsymmetrical P- and N- based nickel complexes combine the unique properties of both classes of ligands. Moreover, POCN complexes turned out to be active catalysts in the processes of functionalization of acrylonitrile and asymmetric Suzuki–Miyaura cross-coupling reaction, while the catalytic efficiency of nickel pincer complexes in the Kharasch addition follows the order: NCN > POC_sp3_OP > PCN > POCN. The utilization of the PCN scaffold can be efficient for the ethylene oligomerization process. The catalytic efficiency of nickel pincer complexes in the cross-coupling of aryl fluorides with aryl Grignard reagents follows the order: NNP > ONP > ONN > NNN, demonstrating the importance of the presence of the P atom in ligand design. Meanwhile, the asymmetry in NNP scaffold causes higher catalytic activity in the polymerization of norbornene compared to symmetrical PNP analogues. Unsymmetrical PNP’ nickel hydrides react with olefins such as ethylene, 1-hexene and norbornene and styrene to generate corresponding Ni–R derivatives. Meanwhile, NHC-based analogues of P-based nickel pincers turned out to be active catalysts in Kumada, Sonogashira and Suzuki–Miyaura cross-coupling reactions, as well as in asymmetric Friedel−Crafts alkylation and transfer hydrogenation of ketones. Unsymmetrical PCP’ scaffold is particularly interesting since complexes based on it have been successfully applied for highly enantioselective catalytic synthesis of P-stereogenic secondary phosphine-boranes.

The steric and electronic effects of ligands have a significant effect on the catalytic properties of the complexes formed by them. Therefore, the design of the ligand frameworks and their combination with the metal play crucial roles in the development of highly active catalysts. In this context, the use of such donor atoms as phosphorus and nitrogen for the development of pincer ligands and nickel complexes, which have the unique properties in both stoichiometric and catalytic reactions, is a very promising task of modern organometallic chemistry. In such species, the difference in the trans effect between the two different donor arms provides the hemilability of the ligand since the group with the weaker trans effect (N) is more likely to dissociate from the metal center, which leads to a vacant coordination site at the metal center and thus can allow for effective coordination and transformation of substrate molecules in the homogeneous catalysis conditions. Thus, phosphorus- and nitrogen-based pincer nickel complexes demonstrated high activities in such processes as the formation of a new carbon–carbon bond (Heck, Suzuki–Miyaura reactions), hydrogenation of aryl and alkyl ketones, alkane dehydrogenation, carboxylation, hydrosilylation, oligo- and polymerization of unsaturated hydrocarbons and others; moreover, unsymmetrical pincer complexes showed higher catalytic activity than their symmetrical analogues. However, to improve the catalytic activity of the complexes bearing unsymmetrical pincer ligands, the strategies for regulating hemilability of coordination site and flexibility of ligand backbone need to be deeply explored.

## Data Availability

Not applicable.

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
