# Peer review of "Recent Advances in Chemistry of Unsymmetrical Phosphorus-Based Pincer Nickel Complexes: From Design to Catalytic Applications"

_molecules, 2021, doi:10.3390/molecules26134063_

Round 1
Reviewer 1 Report
The review article is devoted to the chemistry and catalytic applications of P-based non-symmetric pincer nickel complexes. Considering the current interest in homogenous Ni catalysis and taking into account the lack of revision articles dedicated to this topic, the present review can withdraw the attention of a wide diversity of readers. However, at present I only recommend its publication after major revision.
-Major concerns:
In general, the information provided in the review has to be presented in a more organized manner to make it more attractive and easier to read. I suggest the following changes:
Introduction:
- The authors should better define what a pincer ligand is, emphasizing on the characteristics that make them unique among other chelators.
- References for the first examples of Ni-pincer (PCP and NCN) complexes should be cited in the text.
- I think Fig 2b should be moved to section 2, where the differences on the electronic and steric properties of PCP vs NCN are discussed.
- The information about PCN-type compounds (page 2, from line 57 to 66) should be removed from the introduction and included in the section devoted to PCN complexes (Section 4).
- Figure 2 is not very relevant. It might be eliminated.
- On page 3, in the first paragraph authors report on catalytic properties of metal complexes supported by pincer ligands. This information could be relevant in the context of a general review on metal-pincer complexes but given that this review is focused on Ni-pincer complexes, such information may be briefly summarized although all references can be included in the reference section.
Section 3 (POCN complexes)
- Complex 7 should be included in the same Figure with complexes 8-13
- Page 7, line 155 “…also obtained by the authors…”. Please indicate the author’s name
Section 4 (PCN complexes)
- Page 9, line 196-197, “…are important intermediates of this process”. Which process are the authors referred to?
- Page 10, line 211, “…these authors…” Indicate author’s name
Section 5 (NNP complexes)
- This section is dedicated to NNP ligands and complexes. What is the point of introducing Yamaguchi’s work on NNO, NOP or NNN complexes? It is a bit confusing. Complexes 23-25 should be removed from Scheme 11 and the discussion on Yamaguchi’s work should be focused on the importance of a P group in the structure of the pincer ligand to make the Ni catalyst more active in the Kumada couplings. Then, table 2 is unnecessary.
Section 6 (PCP)
- It should be better name as PCP’
Section 7 (NHC complexes)
- The review is focused on P-based pincer ligands. Hence, none of the examples shown in Schemes 18-23 are representative examples of P-based pincer complexes. They should be removed. The only example provided of P-based complex is 46 (OPCC pincer type). In my opinion, section 7 should be eliminated and complex 46 should be included in the section dedicated to other complexes.
Section 8 (other type complexes)
- Cationic complex 51 is a clear example of PNN pincer complex. It should be included in the section dedicated to PNN complexes.
- Lucani’s compounds 52 and 53 are examples of PCN pincer complexes. Please move them to the corresponding section of the review.
- Zargarian complexes 54a-54f are also PCN type compounds. They should be discussed in the section devoted to PCN complexes.
- The work of Wang should be discussed in the section dedicated to PCP’ complexes.
Author Response
Response to Reviewer 1
Manuscript ID molecules-1257321
Type Review
Title
Recent Advances in Chemistry of Unsymmetrical Phosphorus-Based Pincer Nickel Complexes: From Design to Catalytic Applications
Authors
Zufar Nafigullovich Gafurov * , Artyom O. Kantyukov , Alexey A. Kagilev , Alina A. Kagileva , Il'yas F. Sakhapov , Ilya K. Mikhailov , Dmitry Grigorievich Yakhvarov *
First of all, we would like kindly to thank the reviewers for inspective reading and evaluation of our manuscript.
Reviewer 1.
The review article is devoted to the chemistry and catalytic applications of P-based non-symmetric pincer nickel complexes. Considering the current interest in homogenous Ni catalysis and taking into account the lack of revision articles dedicated to this topic, the present review can withdraw the attention of a wide diversity of readers. However, at present I only recommend its publication after major revision.
Thanks to the Reviewer for the evaluation of our manuscript.
-Major concerns:
In general, the information provided in the review has to be presented in a more organized manner to make it more attractive and easier to read. I suggest the following changes:
Introduction:
The authors should better define what a pincer ligand is, emphasizing on the characteristics that make them unique among other chelators.
Reviewer is right; the corresponding comments have been added into the introduction section of the manuscript according to reviewer’s remark.
References for the first examples of Ni-pincer (PCP and NCN) complexes should be cited in the text.
The corresponding references have been cited in the introduction section of the manuscript according to reviewer’s remark.
I think Fig 2b should be moved to section 2, where the differences on the electronic and steric properties of PCP vs NCN are discussed.
We do believe, that reviewer is writing about Fig. 1b. This figure has been moved to the section 2 according to reviewer’s remark.
The information about PCN-type compounds (page 2, from line 57 to 66) should be removed from the introduction and included in the section devoted to PCN complexes (Section 4).
The corresponding information has been moved to the section 4 according to the reviewer’s remark.
Figure 2 is not very relevant. It might be eliminated.
Reviewer is right; figure 2 has been eliminated from the manuscript according to the reviewer’s remark.
On page 3, in the first paragraph authors report on catalytic properties of metal complexes supported by pincer ligands. This information could be relevant in the context of a general review on metal-pincer complexes but given that this review is focused on Ni-pincer complexes, such information may be briefly summarized although all references can be included in the reference section.
Reviewer is right; catalytic properties of metal complexes supported by pincer ligands is briefly summarized in this paragraph, and we do believe, this information is relevant in the introduction section, since it allows the reader to understand the importance of this type of complexes.
Section 3 (POCN complexes)
Complex 7 should be included in the same Figure with complexes 8-13
Complex 7 was included into the figure with complexes 8-13 according to the reviewer’s remark.
Page 7, line 155 “…also obtained by the authors…”. Please indicate the author’s name
Author’s name [Zargarian] was indicated according to the reviewer’s remark.
Section 4 (PCN complexes)
Page 9, line 196-197, “…are important intermediates of this process”. Which process are the authors referred to?
The process [Kharasch addition reaction] was indicated according to the reviewer’s remark.
Page 10, line 211, “…these authors…” Indicate author’s name.
Author’s name [Moussa] was indicated according to the reviewer’s remark.
Section 5 (NNP complexes)
This section is dedicated to NNP ligands and complexes. What is the point of introducing Yamaguchi’s work on NNO, NOP or NNN complexes? It is a bit confusing. Complexes 23-25 should be removed from Scheme 11 and the discussion on Yamaguchi’s work should be focused on the importance of a P group in the structure of the pincer ligand to make the Ni catalyst more active in the Kumada couplings. Then, table 2 is unnecessary.
Reviewer is right; Yamaguchi’s NNO, NOP or NNN complexes do not belong to the section “NNP complexes”. However, we do believe, that discussion on the way of Yamaguchi’s research, starting from β-aminoketonato and β-diketiminato-based unsymmetrical pincer complexes and explaining the advantages of a P group in the structure of the pincer ligand, which makes the catalyst more active in the Kumada couplings, is justified in the context of the manuscript. Especially, taking into account the title “From Design to Catalytic Applications”. Therefore, the table 2 is also very important to emphasize the advantages of phosphorus-based pincer nickel complexes.
Section 6 (PCP)
It should be better name as PCP’
We do believe, that reviewer is writing about section 6. “PNP complexes”. The section has been renamed to PNP’ according to the reviewer’s remark.
Section 7 (NHC complexes)
The review is focused on P-based pincer ligands. Hence, none of the examples shown in Schemes 18-23 are representative examples of P-based pincer complexes. They should be removed. The only example provided of P-based complex is 46 (OPCC pincer type). In my opinion, section 7 should be eliminated and complex 46 should be included in the section dedicated to other complexes.
Reviewer is right; section 7 represents the examples of nickel pincer complexes bearing the N-Heterocyclic carbene moiety. However, NHCs are isolobal with electron-rich phosphines, they are often considered as phosphine ligand alternatives, therefore, the comparison of synthesis strategies, structural characteristics, and catalytic performance of unsymmetrical pincer nickel complexes bearing NHC groups is justified in the context of the manuscript.
Section 8 (other type complexes)
Cationic complex 51 is a clear example of PNN pincer complex. It should be included in the section dedicated to PNN complexes.
Discussion on complex 51 was moved to the section dedicated to PNN complexes according to the reviewer’s remark.
Lucani’s compounds 52 and 53 are examples of PCN pincer complexes. Please move them to the corresponding section of the review.
Discussion on the complexes 52 and 53 was moved to the section dedicated to PCN complexes according to the reviewer’s remark.
Zargarian complexes 54a-54f are also PCN type compounds. They should be discussed in the section devoted to PCN complexes.
Discussion on the complexes 54a-54f was moved to the section dedicated to PCN complexes according to the reviewer’s remark.
The work of Wang should be discussed in the section dedicated to PCP’ complexes.
Reviewer is right; the work of Wang is dedicated to PCP’ type nickel complexes, however, the manuscript does not contain such a section; therefore, these complexes are discussed in the “other type complexes” section.
Reviewer 2 Report
This is a very good review, which not only classifies the material perfectly, but also allows the reader to understand the importance of this type of complexes. Of course, the reader would like to know the authors' opinion on how these complexes are converted from pre-catalysts to catalysts in the course of the reaction (taking into account the effect on this stage of the induction period) and how the described ligands affect the active catalyst. It should be noted that the review is very easy to read and makes the reader think. In the subject of this review, there is a connection with symmetric and asymmetric NHC-complexes.
Author Response
Response to Reviewer 2
Manuscript ID molecules-1257321
Type Review
Title
Recent Advances in Chemistry of Unsymmetrical Phosphorus-Based Pincer Nickel Complexes: From Design to Catalytic Applications
Authors
Zufar Nafigullovich Gafurov * , Artyom O. Kantyukov , Alexey A. Kagilev , Alina A. Kagileva , Il'yas F. Sakhapov , Ilya K. Mikhailov , Dmitry Grigorievich Yakhvarov *
Reviewer 2.
This is a very good review, which not only classifies the material perfectly, but also allows the reader to understand the importance of this type of complexes. Of course, the reader would like to know the authors' opinion on how these complexes are converted from pre-catalysts to catalysts in the course of the reaction (taking into account the effect on this stage of the induction period) and how the described ligands affect the active catalyst. It should be noted that the review is very easy to read and makes the reader think. In the subject of this review, there is a connection with symmetric and asymmetric NHC-complexes.
Thanks a lot to the Reviewer for the evaluation of our manuscript.
Reviewer 3 Report
The review of Gafurov et al. describes the recent progress in the field of pincer type nickel complexes, their synthesis and applications, exclusively related to their dissymmetrical character. I really enjoyed reading the review which is thorough, supplied by nice graphical figures and the merit within. I believe it will be of interest to people working in catalysis and coordination chemistry. I recommend the article to be published, nonetheless I encourage the authors to make it even better by addressing the questions and thoughts I had while reading the manuscript, which are given below:
- authors write on line 92 'For example, nickel, which is a more affordable low cost and high natural-abundance analogue of platinum and palladium, is known for its low toxicity, and therefore has attracted increasing attention in the catalytic chemistry of transition metals in recent years' - while the part of the cost is accurate, I would be very cautions about writing of low toxicity of nickel compounds. While everything is the matter of dose, many nickel compounds are known to be allergens, even cancirogens so the sentence as it is is not true.
- I notices that for some works the authors present X-ray crystal structures - which I like very much because of the aesthetic appeal but mostly due to the possibility of unambiguous confirmation of the molecular architecture, followed be some structure data such as bond lenths. Authors present some of them (mostly the Ni-P and Ni-N distances) and even construct the table 1 on line 181. I believe there should be one Table that combines not only PCP and NCN complexes, but essentially all of the X-ray determined compounds presented in the paper (grouped within their respective subclasses). This would allow one to more grasp the importance of the molecular design of ligand and the resulting structural features (bonds, angles etc.). What is more, The Ni-halogen or Ni-C distances should also be presented because they are known to affect the catalytical/electrochemical properties.
- What follows the previous comment, the authors are not consequential with their presentation of the X-ray structures. I did not check all of the schemes, but for example molecular representations from ChemDraw presented on Schemes 16,17,21, 22 and 26 can also be depicted as the X-ray structural representations (i.e. the papers that are cited present these data so it should not be the problem). I encourage the authors to do that whenever possible.
- line 488 and Scheme 26; The authors write about bimetallic copper system so it would be good to include the relevant structure in the scheme as well, otherwise it is completely out of context; possibly as the new graph, which shows PONNOP ligand in the middle and diverges into right and left, where on one side there is a bimetallic architecture (Cu as an example) and on the other the synthesized Ni(II) system (with X-ray examples included). This example is particularly interesting.
- Conclusions also could have been more specified in relation to the genres according to which the paper is organized. After reading the conclusions I would like to know what advantages/possibilities there arise if I utilize the PCN scaffold in comparison to the POCN scaffold, or in what field would the PNN scaffold be of interest. The authors write about the importance of the design of the ligand frameworks and their combination with the metal, which play crucial roles in the development of highly active catalysts but it is very general and related only to the N and P donor atoms.
Author Response
Response to Reviewer 3
Manuscript ID molecules-1257321
Type Review
Title
Recent Advances in Chemistry of Unsymmetrical Phosphorus-Based Pincer Nickel Complexes: From Design to Catalytic Applications
Authors
Zufar Nafigullovich Gafurov * , Artyom O. Kantyukov , Alexey A. Kagilev , Alina A. Kagileva , Il'yas F. Sakhapov , Ilya K. Mikhailov , Dmitry Grigorievich Yakhvarov *
First of all, we would like kindly to thank the reviewers for inspective reading and evaluation of our manuscript.
Reviewer 3.
The review of Gafurov et al. describes the recent progress in the field of pincer type nickel complexes, their synthesis and applications, exclusively related to their dissymmetrical character. I really enjoyed reading the review which is thorough, supplied by nice graphical figures and the merit within. I believe it will be of interest to people working in catalysis and coordination chemistry. I recommend the article to be published, nonetheless I encourage the authors to make it even better by addressing the questions and thoughts I had while reading the manuscript, which are given below:
Thanks to the Reviewer for the evaluation of our manuscript.
authors write on line 92 'For example, nickel, which is a more affordable low cost and high natural-abundance analogue of platinum and palladium, is known for its low toxicity, and therefore has attracted increasing attention in the catalytic chemistry of transition metals in recent years' - while the part of the cost is accurate, I would be very cautions about writing of low toxicity of nickel compounds. While everything is the matter of dose, many nickel compounds are known to be allergens, even cancirogens so the sentence as it is is not true.
Reviewer is right; the sentence has been corrected according to the reviewer’s remark.
I notices that for some works the authors present X-ray crystal structures - which I like very much because of the aesthetic appeal but mostly due to the possibility of unambiguous confirmation of the molecular architecture, followed be some structure data such as bond lenths. Authors present some of them (mostly the Ni-P and Ni-N distances) and even construct the table 1 on line 181. I believe there should be one Table that combines not only PCP and NCN complexes, but essentially all of the X-ray determined compounds presented in the paper (grouped within their respective subclasses). This would allow one to more grasp the importance of the molecular design of ligand and the resulting structural features (bonds, angles etc.). What is more, The Ni-halogen or Ni-C distances should also be presented because they are known to affect the catalytical/electrochemical properties.
Thanks to the Reviewer for the remark. The corresponding table (Table A1) was created and added to the Appendix A.
What follows the previous comment, the authors are not consequential with their presentation of the X-ray structures. I did not check all of the schemes, but for example molecular representations from ChemDraw presented on Schemes 16,17,21, 22 and 26 can also be depicted as the X-ray structural representations (i.e. the papers that are cited present these data so it should not be the problem). I encourage the authors to do that whenever possible.
Reviewer is right; corresponding X-ray structural representations were added to the manuscript according to the reviewer’s remark.
line 488 and Scheme 26; The authors write about bimetallic copper system so it would be good to include the relevant structure in the scheme as well, otherwise it is completely out of context; possibly as the new graph, which shows PONNOP ligand in the middle and diverges into right and left, where on one side there is a bimetallic architecture (Cu as an example) and on the other the synthesized Ni(II) system (with X-ray examples included). This example is particularly interesting.
Corresponding graph and X-ray structural representations were added to the manuscript according to the reviewer’s remark.
Conclusions also could have been more specified in relation to the genres according to which the paper is organized. After reading the conclusions I would like to know what advantages/possibilities there arise if I utilize the PCN scaffold in comparison to the POCN scaffold, or in what field would the PNN scaffold be of interest. The authors write about the importance of the design of the ligand frameworks and their combination with the metal, which play crucial roles in the development of highly active catalysts but it is very general and related only to the N and P donor atoms.
Thanks to the Reviewer for the remark. The corresponding discussion has been added to the conclusions according to the reviewer’s remark.
Round 2
Reviewer 1 Report
The authors have fully addressed my comments and I support publication of this manuscript in its current form.